# An Improved Synthetic Method for Sensitive Iodine Containing Tricyclic Flavonoids

**DOI:** 10.3390/molecules27238430

**Published:** 2022-12-02

**Authors:** Mihail Lucian Birsa, Laura G. Sarbu

**Affiliations:** Department of Chemistry, Alexandru Ioan Cuza University of Iasi, 11 Carol I Blvd., 700506 Iasi, Romania

**Keywords:** flavonoids, 1,3-dithiolium salts, dithiocarbamates

## Abstract

The synthesis of new iodine containing synthetic tricyclic flavonoids is reported. Due to the sensitivity of the precursors to the heat and acidic conditions required for the ring closure of the 1,3-dithiolium core, a new cyclization method has been developed. It consists in the treatment of the corresponding iodine-substituted 3-dithiocarbamic flavonoids with a 1:1 (*v*/*v*) mixture of glacial acetic acid–concentrated sulfuric acid at 40 °C. The synthesis of the iodine-substituted 3-dithiocarbamic flavonoids has also been tuned in terms of reaction conditions.

## 1. Introduction

Flavonoids are a diverse group of polyphenolic plant secondary metabolites. Associated with the multitude of substitution patterns on the C-6–C-3–C-6 backbone, more than 9000 flavonoids are known [1]. The attention that they receive is a direct consequence of the many biological activities that this class of compounds displays. Studies performed on flavonoids found that they possess antioxidant, anti-inflammatory, antimicrobial, antitumoral, antiviral or cardioprotective properties [2,3,4]. The antimicrobial properties that some flavonoids display could be exploited for this purpose. In principle, flavonoids can act directly against the infectious microorganisms, they can be used in combination with other antibiotics (synergistic relationship), or they can act against bacterial virulence factors, such as the cell-binding ability or toxins released by the pathogens. Many flavonoids, such as quercetin and naringenin [5], apigenin [6] or epigallocathechin gallate [7], to name but a few, are known to possess antibacterial activity. More than that, epigallocathechin gallate was also shown to enhance the activity of other antibiotics against drug-resistant pathogens [8]. In the past few years, the subject of antibacterial research has often been related to semisynthetic and synthetic flavonoids, some of these compounds being more active than natural flavonoids [9]. Our recent review highlighted the synthetic flavonoids with antimicrobial activities known up to date in the literature [10].

The emergence of more and more nosocomial infections caused by multidrug-resistant organisms (MDROs) is one of the most worrying phenomena of recent years. The discovery of new and more efficient antimicrobial drugs is therefore a matter of high priority among scientists and clinicians worldwide. Ideally, antibacterial agents should belong to new classes, since the structural alteration of drugs to which resistance has already developed rarely provides a major solution [11]. Following the general interest for synthetic flavonoids, the synthesis of a new class of tricyclic flavonoids as a combination of a condensed benzopyran core and 1,3-dithiolium ring was reported [12]. Subsequently, this class of new synthetic flavonoids proved to exhibit good to excellent antibacterial activities against both Gram-positive and Gram-negative bacteria [13]. The tricyclic flavonoids developed by us inhibited and also killed bacterial cells at very low concentrations (up to 0.24 µg/mL MIC and MBC values) [14,15]. Moreover, some of these flavonoids exhibited a stronger inhibitory and bactericidal effect compared with some antibiotics and other natural or synthetic flavonoids reported in the literature and inhibited to some degree the proliferation of cancer cells [16].

Recently, we reported a study on the influence of halogen substituents on the antibacterial properties of tricyclic flavonoids [17]. Upon going from fluorine to iodine, these compounds exhibited good to excellent antimicrobial properties against both Gram-positive and Gram-negative pathogens. The results suggested that halogen size was the main factor for the change in potency rather than polarity/electronics. Prompted by these findings, we decided to investigate the synthesis of sulphur-containing tricyclic flavonoids bearing two iodine substituents on the benzopyran moiety.

## 2. Results and Discussion

The synthetic route used to obtain 1,3-dithiolium flavonoids **5a**–**e** is described in Figure 1 and follows the protocol used for the model compound **5a**. 2-Bromo-1-(2-hydroxy-3,5-diiodophenyl)ethan-1-one (**1**) [18] readily underwent nucleophilic substitution in the presence of the *N,N*-diethyldithiocarbamate anion, in acetone, yielding the desired phenacyl carbodithioate **2**. The incorporation of the *N,N*-diethyldithiocarbamic unit was confirmed by NMR spectral data. Thus, the ^1^H NMR spectrum indicated the presence of two triplets, at 1.30 ppm and 1.38 ppm, corresponding to the two methyl groups, and also two quartets, 3.83 ppm and 4.03 ppm, provided by the two methylene units directly bound to the nitrogen atom. The ^13^C NMR spectrum confirmed the presence of the two methyl groups (11.5 ppm and 12.6 ppm), the two nitrogen-bounded methylene groups (47.3 ppm and 50.5 ppm) and the thiocarbonyl carbon atom (192.8 ppm).

The reaction of 1-(2-hydroxy-3,5-diiodophenyl)-1-oxa-ethan-2-yl *N,N*-diethylaminocarbodithioate (**2**) with aminals **3** provided 3-substituted dithiocarbamic flavanones **4a**–**e** as a mixture of diastereoisomers (Figure 1). Aminals **3** were synthesized according to the literature procedures [19,20]. Due to the low solubility of dithiocarbamate **2** in ethanol, an improved experimental procedure using a mixture of chloroform and methanol (1:1) as solvent was developed. Thus, the homogeneous reaction mixture was heated at reflux for 4 h. After cooling, pale yellow precipitates were formed that were filtered, dried and recrystallized from ethanol to provide 3-dithiocarbamic flavanones **4a**–**e**, as an inseparable mixture of diastereoisomers, in 68–80% yields. NMR spectra supported the benzopyran ring closure. Thus, besides the NMR pattern of *para*-substituted aromatic ring originating from aminal **3**, we observed the disappearance of the signal of the methylene group from dithiocarbamate **2** (4.86 ppm) and the presence of the characteristic pattern of vicinal hydrogen atoms at the C-2 and C-3 positions of the benzopyran ring for both diastereoisomers between 5.7 and 6 ppm. Because these two protons can be located either on the same side or on opposite sides of the plane of the molecule, two stereoisomers, *anti*-**4**′ and *syn*-**4**″ can be obtained (Figure 1). The relative orientation of the two hydrogen atoms would, of course, be expected to have an influence on the magnitude of their coupling constants. The *anti* isomers always displayed a coupling constant between 6.2 and 7.3 Hz and the *syn* isomers around 4 Hz. The coupling constants and diastereoisomeric ratios of flavonoids **4a**–**e** are presented in Table 1. A ^13^C NMR analysis confirmed the presence of the C-2 carbon atom, found around 80.0 ppm, while the C-3 carbon atom could be found around 60.0 ppm.

α-Ketodithiocarbamates are valuable precursors for 2-dialkylamino-1,3-dithiolium-2-yl cations [21,22,23]. Usually, the acid-catalysed cyclocondensation of these substrates is the method employed for the synthesis of the desired 1,3-dithiolium cations. This consisted in using a glacial acetic acid/sulfuric acid 3:1 (*v*/*v*) at 80 °C for 10 min [24]. Previously, we developed specific methods for the sensitive starting materials prone to decomposition under regular reaction conditions. In one such application, a mixture of phosphorus pentoxide–methanesulfonic acid 1:10 (*w*/*v*) was used for the synthesis of several 4-iodoaryl-1,3-dithiolium salts [25].

Despite our previous experience with the synthesis of tricyclic flavonoids of type **5** [13,14,17], attempts to close the 1,3-dithiolium ring on flavanones **4** led to a black intractable material. Even under mild reaction conditions described by us for iodine-substituted phenacyl dithiocarbamates [25,26], the cyclization reactions failed for all new reported flavanones **4**. Consequently, we tuned the reaction conditions in terms of reducing the reaction temperature and the composition of the cyclization mixture. The best results for our substrates were obtained using a mixture of glacial acetic acid/sulfuric acid 1:1 (*v*/*v*) at 40 °C for 30 min, followed by a treatment with an aqueous solution of sodium tetrafluoroborate. Thus, the tricyclic 1,3-dithiolium flavonoids **5** was obtained as white crystals in 80–88% yields. 

The cyclization of dithiocarbamates **4** to tricyclic flavonoids **5** was accompanied by important spectral changes. Thus, IR spectroscopy showed the absence of the carbonyl absorption bands (1690–1700 cm^−1^) and the presence of new strong and broad absorption bands (ca. 1070 cm^−1^) from the tetrafluoroborate anion. In the ^1^H NMR spectra, the doublets corresponding to the C-3 hydrogens disappeared; at the same time, the signals of the C-2 hydrogens were shifted to ca. 6.9 ppm and became singlets. The ^13^C NMR spectra confirmed the absence of the carbonyl and thiocarbonyl atoms and showed a new signal at ca. 185 ppm corresponding to the 1,3-dithiol-2-ylium carbon atom. 

## 3. Materials and Methods

### 3.1. Chemistry

Melting points were obtained on a *KSPI* melting-point meter (A. KRÜSS Optronic, Hamburg, Germany) and were uncorrected. IR spectra were recorded on a Bruker Tensor 27 instrument (Bruker Optik GmbH, Ettlingen, Germany). NMR spectra were recorded on a Bruker 500 MHz spectrometer (Bruker BioSpin, Rheinstetten, Germany). Chemical shifts are reported in ppm downfield from TMS. UV–vis spectra were recorded on a Varian BioChem 100 spectrophotometer. Mass spectra were recorded on a Thermo Scientific ISQ LT instrument (Thermo Fisher Scientific Inc., Waltham, MA, USA). All reagents were commercially available and used without further purification. Elemental analysis, nuclear magnetic resonance data and copies of ^13^C-NMR spectra are included in the Appendix A.

#### 3.1.1. 1-(2-Hydroxy-3,5-diiodophenyl)-1-oxoethan-2-yl *N*,*N*-diethylamino-1-carbodithioate (**2**)

To a solution of 2-bromo-1-(2-hydroxy-3,5-diiodophenyl)ethan-1-one (**1**, 1.4 g, 3 mmol) in acetone (10 mL), a solution of sodium *N*,*N*-diethyldithiocarbamate trihydrate (0.68 g, 3 mmol) in acetone/water (10 mL, 1:1 *v*/*v*) was added. The resulting mixture was refluxed for 10 min, cooled to room temperature and poured into water (100 mL) with vigorous stirring. The precipitate thus formed was vacuum-filtered and recrystallized from ethanol, yielding 1.3 g (81%) of yellow crystals; M.p. = 162–163 °C. IR (ATR, cm^−1^) 1699, 1499, 1425, 1245, 1174, 821, 621. ^1^H NMR (CDCl_3_) δ 12.71 (s, 1H), 8.30 (d, *J* = 1.6 Hz, 1H), 8.25 (d, *J* = 1.7 Hz, 1H), 4.86 (s, 2H), 4.03 (q, *J* = 6.9 Hz, 2H), 3.83 (q, *J* = 6.9 Hz, 2H), 1.38 (t, *J* = 6.9 Hz, 3H), 1.30 (t, *J* = 6.9 Hz, 3H). ^13^C NMR (CDCl_3_) δ 198.1, 192.8, 160.6, 152.9, 138.7, 120.6, 88.2, 80.6, 50.5, 47.3, 43.5, 12.6, 11.5. UV–vis (λ_max_, nm) 373. MS (EI) (*m*/*z*): 534.8 (M^+^, 37%) for C_13_H_15_I_2_NO_2_S_2_.

#### 3.1.2. General Procedure for 6,8-Diiodo-2-(4-methylphenyl)-4-oxochroman-3-yl *N*,*N*-diethyldithiocarbamate (**4a**)

To a solution of 1-(3,5-diiodo-2-hydroxyphenyl)-1-oxoethan-2-yl *N*,*N*-diethyldithiocarbamate (**2**) (0.268 g, 0.5 mmol) in a mixture of CHCl_3_/MeOH (12 mL, 1:1 *v*/*v*) aminal **3a** (0.13 g, 0.5 mmol) was added and the reaction mixture was heated under reflux for 4 h. After cooling, the solid material was filtered off and purified by recrystallization from ethanol to give **4a** (0.23 g, 72%) as colourless crystals. IR (ATR, cm^−1^) 2738, 1698, 1419, 1255, 1203, 963, 811, 506, 485, 430. ^1^H NMR (CDCl_3_, selected data for the major isomer) δ 8.26 (d, *J* = 1.7 Hz, 1H), 8.13 (d, *J* = 1.7 Hz, 1H), 7.38 (d, *J* = 7.7 Hz, 2H), 7.16 (d, *J* = 7.7 Hz, 2H), 6.01 (d, *J* = 6.2 Hz, 1H), 5.75 (d, *J* = 6.2 Hz, 1H), 3.99 (m, 2H), 3.68 (m, 2H), 2.35 (s, 3H), 1.25 (t, *J* = 6.9 Hz, 6H). ^13^C NMR (CDCl_3_, selected data for the major isomer) δ 191.6, 186.2, 158.8, 152.6, 138.7, 136.3, 132.8, 129.3, 127.2, 122.9, 87.6, 84.7, 83.0, 57.8, 50.5, 47.3, 21.2, 12.6, 11.4. UV–vis (λ_max_, nm) 314. MS (EI) *m*/*z*: 636.8 (M^+^, 17%) for C_21_H_21_I_2_NO_2_S_2_.

#### 3.1.3. 6,8-Diiodo-2-(4-ethylphenyl)-4-oxochroman-3-yl *N*,*N*-diethyldithiocarbamate (**4b**)

Colourless crystals, 0.22 g, 68%. IR (ATR, cm^−1^) 2965, 1697, 1417, 1256, 1202, 825, 641, 474, 438. ^1^H NMR (CDCl_3_, selected data for the major isomer) δ 8.26 (d, *J* = 1.8 Hz, 1H), 8.13 (d, *J* = 1.8 Hz, 1H), 7.4 (d, *J* = 7.9 Hz, 2H), 7.19 (d, *J* = 7.9 Hz, 2H), 6.01 (d, *J* = 6.3 Hz, 1H), 5.76 (d, *J* = 6.3 Hz, 1H), 3.98 (m, 2H), 3.69 (m, 2H), 2.65 (q, *J* = 7.5 Hz, 2H), 1.25 (t, *J* = 7.5 Hz, 3H), 1.23 (t, *J* = 7.6 Hz, 6H). ^13^C NMR (CDCl_3_, selected data for the major isomer) δ 191.2, 186.3, 158.8, 152.6, 144.9, 136.3, 133.0, 128.1, 127.2, 122.9, 87.6, 84.7, 83.0, 57.8, 50.5, 47.3, 28.5, 15.3, 12.6, 11.4. UV–vis (λ_max_, nm) 313. MS (EI) *m*/*z*: 650.8 (M^+^, 27%) for C_22_H_23_I_2_NO_2_S_2_.

#### 3.1.4. 6,8-Diiodo-2-(4-fluorophenyl)-4-oxochroman-3-yl *N*,*N*-diethyldithiocarbamate (**4c**)

Colourless crystals, 0.24 g, 75%. IR (ATR, cm^−1^) 2980, 1685, 1419, 1226, 1201, 974, 825, 537, 474. ^1^H NMR (CDCl_3_, selected data for the major isomer) δ 8.27 (d, *J* = 1.9 Hz, 1H), 8.16 (d, *J* = 1.9 Hz, 1H), 7.49 (m, 2H), 7.06 (m, 2H), 5.96 (d, *J* = 7.3 Hz, 1H), 5.79 (d, *J* = 7.3 Hz, 1H), 3.95 (m, 2H), 3.71 (m, 2H), 1.24 (t, *J* = 6.8 Hz, 6H). ^13^C NMR (CDCl_3_, selected data for the major isomer) δ 190.9, 186.1, 162.9, 158.7, 152.7, 136.4, 131.7, 129.4, 122.4, 115.5, 87.4, 84.9, 82.6, 58.4, 50.7, 47.3, 12.6, 11.4. UV–vis (λ_max_, nm) 308. MS (EI) *m*/*z*: 640.8 (M^+^, 24%) for C_20_H_18_FI_2_NO_2_S_2_.

#### 3.1.5. 6,8-Diiodo-2-(4-bromophenyl)-4-oxochroman-3-yl *N*,*N*-diethyldithiocarbamate (**4d**)

Colourless crystals, 0.28 g, 80%. IR (ATR, cm^−1^) 2977, 1700, 1421, 1252, 1201, 817, 507, 433, 417. ^1^H NMR (CDCl_3_, selected data for the major isomer) δ 8.28 (d, *J* = 1.9 Hz, 1H), 8.16 (d, *J* = 1.9 Hz, 1H), 7.52 (d, *J* = 8.2 Hz, 2H), 7.41 (d, *J* = 8.2 Hz, 2H), 5.95 (d, *J* = 7.1 Hz, 1H), 5.74 (d, *J* = 7.1, 1H), 3.92 (m, 2H), 3.66 (m, 2H), 1.24 (t, *J* = 6.7 Hz, 6H). ^13^C NMR (CDCl_3_, selected data for the major isomer) δ 190.8, 185.9, 158.7, 152.7, 136.5, 134.8, 131.7, 129.1, 123.1, 122.1, 87.4, 85.1, 82.6, 58.2, 50.7, 47.4, 12.6, 11.4. UV–vis (λ_max_, nm) 311. MS (EI) *m*/*z*: 700.7 (M^+^, 296%) for C_20_H_18_BrI_2_NO_2_S_2_.

#### 3.1.6. 6,8-Diiodo-2-(4-methoxyphenyl)-4-oxochroman-3-yl *N*,*N*-diethyldithiocarbamate (**4e**)

Colourless crystals, 0.25 g, 77%. IR (ATR, cm^−1^) 1699, 1496, 1420, 1246, 1180, 1032, 828, 663. ^1^H NMR (CDCl_3_, selected data for the major isomer) δ 8.26 (m, 1H), 8.14 (m, 1H), 7.42 (d, *J* = 8.2 Hz, 2H), 6.89 (d, *J* = 8.2 Hz, 2H), 5.96 (d, *J* = 6.7 Hz, 1H), 5.78 (d, *J* = 6.7 Hz, 1H), 3.97 (m, 2H), 3.81 (s, 3H), 3.65 (m, 2H), 1.25 (t, *J* = 6.7 Hz, 6H). ^13^C NMR (CDCl_3_, selected data for the major isomer) δ 191.2, 186.4, 159.9, 158.8, 152.6, 136.4, 128.7, 127.9, 122.8, 113.9, 87.7, 87.6, 84.7, 58.1, 55.3, 50.5, 47.3, 12.6, 11.4. UV–vis (λ_max_, nm) 315. MS (EI) *m*/*z*: 652.8 (M^+^, 37%) for C_21_H_21_I_2_NO_3_S_2_.

#### 3.1.7. General Procedure for 2-*N*,*N*-diethylamino-6,8-diiodo-4-(4-methylphenyl)-*4H*-1,3-dithiol[4,5-*c*]chromen-2-ylium tetrafluoroborate (**5a**)

To a mixture of sulfuric acid (1 mL) and acetic acid (1 mL), flavanone **4a** (0.21 g, 0.33 mmol) was added and the resulting solution was heated to 40 °C for 30 min. The reaction mixture was then left to cool to room temperature and a solution of sodium tetrafluoroborate (0.2 g) in water (10 mL) was added dropwise, with vigorous stirring. The resulting precipitate was then filtered, washed thoroughly with water and recrystallized from ethanol, yielding the desired tetrafluoroborate **5a** in the form of colourless crystals (0.17 g, 85%). M.p. 260–261 °C. IR (ATR, cm^−1^) 1554, 1438, 1224, 1045, 729, 494, 458. ^1^H NMR (DMSO-*d*6) δ 8.04 (d, *J* = 1.8 Hz, 1H), 7.74 (d, *J* = 1.8 Hz, 1H), 7.37 (d, *J* = 8.0 Hz, 2H), 7.26 (d, *J* = 8.0 Hz, 2H), 6.89 (s, 1H), 3.89 (m, 4H), 2.31 (s, 3H), 1.40 (t, *J* = 7.1 Hz, 3H), 1.32 (t, *J* = 7.1 Hz, 3H). ^13^C NMR (DMSO-*d*6) δ 184.9, 150.7, 147.8, 140.2, 133.8, 130.0, 129.7, 127.8, 126.8, 119.1, 88.7, 87.6, 76.2, 54.7, 54.6, 21.3, 10.8, 10.5. UV–vis (λ_max_, nm) 343. MS (EI) *m*/*z*: 619.9 (M^+^-BF_4,_ 7%) for C_21_H_20_I_2_NOS_2_]^+^.

#### 3.1.8. 2-*N*,*N*-Diethylamino-6,8-diiodo-4-(4-ethylphenyl)-*4H*-1,3-dithiol[4,5-*c*]chromen-2-ylium tetrafluoroborate (**5b**)

Colourless crystals, M.p. 201–202 °C, (0.17 g, 81%). IR (ATR, cm^−1^) 1549, 1428, 1217, 1034, 719, 496, 448. ^1^H NMR (DMSO-*d*6) δ 8.06 (d, *J* = 1.8 Hz, 1H), 7.73 (d, *J* = 1.8 Hz, 1H), 7.35 (d, *J* = 8.1 Hz, 2H), 7.24 (d, *J* = 8.1 Hz, 2H), 6.87 (s, 1H), 3.87 (m, 4H), 2.35 (q, *J* = 7.3 Hz, 2H), 1.40 (t, *J* = 7.1 Hz, 3H), 1.32 (t, *J* = 7.1 Hz, 3H), 1.26 (t, *J* = 7.3 Hz, 3H). ^13^C NMR (DMSO-*d*6) δ 185, 150.5, 147.7, 140.1, 133.5, 129.9, 129.6, 127.5, 126.7, 119.0, 88.8, 87.5, 76.1, 54.8, 54.6, 25.4, 12.3, 10.7, 10.4. UV–vis (λ_max_, nm) 344. MS (EI) *m*/*z*: 633.9 (M^+^-BF_4_, 5%) for C_22_H_22_I_2_NOS_2_]^+^.

#### 3.1.9. 2-*N*,*N*-diethylamino-6,8-diiodo-4-(4-fluorophenyl)-*4H*-1,3-dithiol[4,5-*c*]chromen-2-ylium tetrafluoroborate (**5c**)

Colourless crystals, M.p. 237–238 °C (0.17 g, 83%). IR (ATR, cm^−1^) 1551, 1433, 1225, 1048, 685, 458, 409. ^1^H NMR (DMSO-*d*6) δ 8.09 (d, *J* = 1.8 Hz, 1H), 7.80 (d, *J* = 1.8 Hz, 1H), 7.55 (dd, ^3^*J*_H-H_ = 8.7 Hz, ^4^*J*_H-F_ = 5.3 Hz, 2H), 7.30 (dd, ^3^*J*_H-H_ = 8.8 Hz, ^3^*J*_H-F_ = 8.7 Hz, 2H), 6.96 (s, 1H), 3.90 (m, 4H), 1.40 (t, *J* = 7.1 Hz, 3H), 1.33 (t, *J* = 7.1 Hz, 3H). ^13^C NMR (DMSO-*d*6) δ 185.0, 164.2, 162.3, 150.4, 147.8, 133.0, 130.3, 129.2, 127.1, 119.0, 116.5, 88.7, 87.7, 75.5, 54.7, 54.6, 10.8, 10.5. UV–vis (λ_max_, nm) 339. MS (EI) *m*/*z*: 623.8 (M^+^-BF_4_, 8%) for C_20_H_17_FI_2_NOS_2_]^+^.

#### 3.1.10. 2-*N*,*N*-diethylamino-6,8-diiodo-4-(4-bromophenyl)-*4H*-1,3-dithiol[4,5-*c*]chromen-2-ylium tetrafluoroborate (**5d**)

Colourless crystals, M.p. 219–220 °C (0.2 g, 88%). IR (ATR, cm^−1^) 1546, 1429, 1225, 1049, 737, 441, 428. ^1^H NMR (DMSO-*d*6) δ 8.09 (d, *J* = 1.7 Hz, 1H), 7.79 (d, *J* = 1.7 Hz, 1H), 7.66 (d, *J* = 8.4 Hz, 2H), 7.43 (d, *J* = 8.4 Hz, 2H), 6.92 (s, 1H), 3.87 (m, 4H), 1.39 (t, *J* = 7.1 Hz, 3H), 1.32 (t, *J* = 7.1 Hz, 3H). ^13^C NMR (DMSO-*d*6) δ 185.0, 150.4, 147.9, 136.0, 133.1, 132.5, 130.0, 128.7, 127.2, 123.9, 119.0, 88.6, 87.8, 75.5, 54.7, 54.6, 10.7, 10.5. UV–vis (λ_max_, nm) 341. MS (EI) *m*/*z*: 683.7 (M^+^-BF_4_, 5%) for C_20_H_17_BrI_2_NOS_2_]^+^.

#### 3.1.11. 2-*N*,*N*-diethylamino-6,8-diiodo-4-(4-methoxyphenyl)-*4H*-1,3-dithiol[4,5-*c*]chromen-2-ylium tetrafluoroborate (**5e**)

Colourless crystals, M.p. 235–236 °C (0.17 g, 80%). IR (ATR, cm^−1^) 1548, 1429, 1247, 1070, 851, 617. ^1^H NMR (DMSO-*d*6) δ 8.08 (d, *J* = 1.8 Hz, 1H), 7.78 (d, *J* = 1.8 Hz, 1H), 7.41 (d, *J* = 8.7 Hz, 2H), 6.99 (d, *J* = 8.7 Hz, 2H), 6.86 (s, 1H), 3.90 (m, 4H), 3.76 (s, 3H), 1.40 (t, *J* = 7.1 Hz, 3H), 1.32 (t, *J* = 7.1 Hz, 3H). ^13^C NMR (DMSO-*d*6) δ 184.9, 160.9, 150.7, 147.7, 132.9, 129.9, 129.6, 128.6, 126.7, 119.0, 114.8, 88.7, 87.4, 76.1, 55.7, 54.7, 54.6, 10.8, 10.5. UV–vis (λ_max_, nm) 342. MS (EI) *m*/*z*: 635.8 (M^+^-BF_4_, 9%) for C_21_H_20_I_2_NO_2_S_2_]^+^.

## 4. Conclusions

In conclusion, we reported the synthesis of five iodine-containing tricyclic flavonoids, whose backbone is known to induce antimicrobial properties. This was performed through a new synthetic approach using a glacial acetic acid/sulfuric acid 1:1 (*v*/*v*) mixture at 40 °C as a cyclization agent. The synthesis of the precursors 3-dithiocarbamic flavanone was also tuned in terms of the reaction conditions.

## Data Availability

Not applicable.

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
