# Peer review of "An Improved Synthetic Method for Sensitive Iodine Containing Tricyclic Flavonoids"

_molecules, 2022, doi:10.3390/molecules27238430_

Round 1

Reviewer 1 Report

The authors have tried to report the synthesis of new iodine containing synthetic tricyclic flavonoids, the results is interestingly. I recommend this work could be accepted based on the below points.

1.       Check the typesetting errors carefully, such as C6-C3-C6 backbone,

2.       Compared similar report on this methods.

3.       Some similar work on this topic may be considered, such as Org. Chem. Front., 2018, 5, 653–661;J. Org. Chem. 2019, 84, 14627−14635; Org. Lett. 2020, 22, 8086−8090; Polym. Chem., 2022, 13, 2351–2361 and Chem. Commun., 2022, 58, 6653–6656

4.       List a Table for 1H and 13C NMR data of the synthesized compounds.

5.       Please provide Uv–vis spectra of the oligomers

Author Response

Dear Reviewer,

Thank you for your observation. Please find below the answers.

  1. Check the typesetting errors carefully, such as C6-C3-C6 backbone.

Done

  1. Compared similar report on this methods.

A sentence about a previous method for the ring closure of the 1,3-dithiolium core has been added, lines 99, 100. As we mentioned in the manuscript attempts to close the 1,3-dithiolium ring on flavanones 4 using conventional methods led to a black intractable material.

  1. Some similar work on this topic may be considered, such as Org. Chem. Front., 2018, 5, 653–661;J. Org. Chem. 2019, 84, 14627−14635; Org. Lett. 2020, 22, 8086−8090; Polym. Chem., 2022, 13, 2351–2361 and Chem. Commun., 2022, 58, 6653–6656

      As suggested, two relevant similar work on this topic have been cited (references 22 and 23)

  1. List a Table for 1H and 13C NMR data of the synthesized compounds.

A table for 1H and 13C NMR data of the synthesized compounds has been added in the supplementary material (Table S2)

  1. Please provide Uv–vis spectra of the oligomers.

Done

Reviewer 2 Report

In this article, the authors present a new methodology for the synthesis of tricyclic flavonoids containing sensitive iodine units.

After a good bibliographic introduction, showing the interest of this class of molecules, the authors explain their strategy to develop a new family of compounds containing iodine functions, as their presence plays an important role for biological applications.

The results and discussion are easy to understand, the spectroscopic information demonstrating the formation of these new molecules are clearly given. I would like to emphasise the quality of the experimental results and the 13C NMR spectra.

I think this article will be interesting for the readers of "Molecules", so I recommend acceptance of this article with minor corrections:

-In the title, the misspelling of the word "tricyclic" should be corrected.

- On page 4, l120, the signal is 185 ppm and not 189 ppm (for 5a), the same error should be corrected also on page 5, line 209 (cf 13C NMR).  In the same sentence, the term "positive" does not make sense and can be deleted.

- for the mixture of diastereomers 4a-4e, I think that the melting points are unnecessary (as it is a mixture of compounds and not pure stereoisomers).

- on page 6 l241, the number 79 before the bromine atom should be deleted.

Author Response

Dear Reviewer,

Thank you for your pertinent observations and please accept our apologize for the "tryciclic" (in title).

All of your observations have been corrected.

Best wishes.